# Smoking History and Nicotine Dependence Alter Sleep Features in Patients with Obstructive Sleep Apnea-Hypopnea Syndrome

**DOI:** 10.3390/healthcare13010049

**Published:** 2024-12-30

**Authors:** Ioanna Grigoriou, Serafeim-Chrysovalantis Kotoulas, Konstantinos Porpodis, Dionysios Spyratos, Ioanna Papagiouvanni, Alexandros Tsantos, Anastasia Michailidou, Constantinos Mourelatos, Christina Mouratidou, Ioannis Alevroudis, Kalliopi Tsakiri, Vasiliki Dourliou, Agni Sakkou, Sotirios Matzolas, Alexandra Marneri, Athanasia Pataka

**Affiliations:** 1Respiratory Failure Clinic and Sleep Laboratory, General Hospital of Thessaloniki “G. Papanikolaou”, Aristotle’s University of Thessaloniki, 541 24 Thessaloniki, Greece; ioagrig@hotmail.gr (I.G.);; 2Adult ICU, General Hospital of Thessaloniki “Ippokrateio”, 546 42 Thessaloniki, Greece; 3Pulmonary Department, General Hospital of Thessaloniki “G. Papanikolaou”, Aristotle’s University of Thessaloniki, 541 24 Thessaloniki, Greece; 44th Internal Medicine Department, General Hospital of Thessaloniki “Ippokrateio”, Aristotle’s University of Thessaloniki, 546 42 Thessaloniki, Greece; 5Pulmonary Department General, Hospital of Thessaloniki “Ippokrateio”, 546 42 Thessaloniki, Greece; 62nd Propaedeutic Internal Medicine Department, General Hospital of Thessaloniki “Ippokrateio”, Aristotle’s University of Thessaloniki, 541 24 Thessaloniki, Greece; 7Genetics Laboratory, Aristotle’s University of Thessaloniki, 541 24 Thessaloniki, Greece

**Keywords:** obstructive sleep apnea-hypopnea syndrome, smoking history, nicotine dependance, sleep disorders

## Abstract

**Introduction:** There are many aspects in the relationship between smoking and sleep that have not been investigated thoroughly yet, especially in regards to obstructive sleep apnea-hypopnea syndrome (OSAHS). **Methods:** In this cross-sectional study, 2359 participants, who have visited the sleep clinic of our hospital during a 13-year period and were former or current smokers, were included. Their smoking history, measured in packyears of smoking, and their nicotine dependence, measured with the Fagerström scale, were correlated with various epidemiological and sleep-related variables. **Results:** Patients with respiratory, cardiovascular and metabolic comorbidities were older, more obese and presented a significantly greater history in packyears of smoking. Packyears were positively correlated with the Epworth sleepiness scale (ESS) (r = 0.06, *p* = 0.007), with %REM sleep time (r = 0.19, *p* = 0.042), apnea-hypopnea index (AHI) (r = 0.10, *p* < 0.001), oxygen desaturation index (ODI) (r = 0.10, *p* < 0.001), mean and maximum apnea duration (r = 0.10, *p* < 0.001 and r = 0.11, *p* < 0.001, respectively), while they were negatively correlated with mean and minimum SaO_2_ (r = −0.18, *p* < 0.001 and r = −0.13, *p* < 0.001, respectively). Furthermore, smoking history exhibited a significantly increasing trend with increasing OSA diagnosis and severity (*p* < 0.001). Patients with abnormal movements during sleep and those with restless sleep showed a significantly higher nicotine dependence, measured with the Fagerström scale, compared to those without abnormal movements or restless sleep (5.4 ± 2.8 vs. 4.7 ± 2.8, *p* = 0.002 and 5.1 ± 2.9 vs. 4.7 ± 2.7, *p* = 0.043). **Conclusions:** Smoking history in packyears probably affects OSAHS characteristics, while nicotine dependence seems to be related more with abnormal sleep behaviors.

## 1. Introduction

According to the World Health Organization (WHO), smoking is a chronic, relapsing and difficult to treat disease, responsible for increased healthcare-related cost, morbidity and mortality [1]. Smoking is one of the major risk factors for ischemic heart and cerebral diseases and chronic obstructive pulmonary disease (COPD) [2], while these diseases are categorized among the leading causes of death worldwide [3].

The prevalence of obstructive sleep apnea-hypopnea syndrome (OSAHS) varies between 3% and 7% in the general population [4]. Apart from excessive daytime sleepiness, other OSAHS symptoms include snoring, gasping or choking episodes during sleep, frequent awakenings and non-refreshing sleep in general [4]. Apneic events might emerge at every sleep stage; however, they are more prevalent in rapid eye movement (REM) sleep because in that stage muscle tone diminishes [4]. OSA is also considered a risk factor for cardiovascular disease. The dynamic narrowing of upper airways during sleep increases blood pressure and causes significant intrathoracic pressure swings, which are considered as the main pathophysiologic mechanisms that increase the risk for cardiovascular events in OSA patients [5]. Furthermore, apneic episodes during sleep, apart from hypertension, also cause increased sympathetic tone, hypoxia and hypercarbia, leading to endothelial dysfunction [6].

Since both smoking and OSA are risk factors for cardiovascular diseases, many studies have sought to investigate for a possible relationship between them. Smoking shows a slightly higher prevalence in patients with OSAHS compared to the general population [7,8]. Neuromuscular dysfunction and inflammation enhancement of the upper airway, which provoke its collapse during sleep, and frequent awakenings and sleep architecture fragmentation, which exacerbate daytime sleepiness, have all been suggested as possible pathophysiological mechanisms through which smoking affects OSAHS [9]. Active or passive smoking and smoking history have been linked with snoring [10]. Smoking might contribute to OSAHS symptoms through worsening chronic airway inflammation [11], promoting snoring and apneas [9,10,12]. Additionally, active smokers exhibit poor sleep quality with difficulty in falling asleep and maintaining sleep [13,14,15]. Yet, there is still need for more studies which investigate the relationship between smoking and OSAHS, since evidence is still conflicting.

Apart from OSAHS, other sleep disorders, such as sleeptalking, sleepwalking, sleep-related eating disorder, bruxism, abnormal movements during sleep, restless legs, nightmares, night terrors and sleep paralysis, have also been related with smoking in the form of passive smoking exposure during pregnancy or early childhood [16,17]. Furthermore, sleep paralysis, sleep-related eating disorder and REM behavior disorder (RBD) have been associated with smoking directly [18,19,20,21], although another study did not find any relationship between parasomnias and smoking [22].

Although there are several studies that have examined for a possible relationship between smoking and sleep, this relationship is not quite clear yet. More importantly, there are very few studies that have tried to quantify the relationship between those two. The aim of this study was to investigate for a possible quantitative relationship between smoking history and nicotine dependence on the one hand, and OSAHS or other sleep disorders on the other, by using the packyears of smoking and the Fagerström scale as quantitative indices for smoking history and nicotine dependence, respectively.

## 2. Methods

### 2.1. Ethics

This cross-sectional study was approved by the ethics committee of the medical school of the Aristotle’s University of Thessaloniki, Greece (https://www.med.auth.gr/ (accessed on 30 September 2024)), protocol number: 47/2022, on 14 December 2022. All participants provided a written informed consent, along with the questionnaires that were used for this study.

### 2.2. Population

The records of all adult patients who have visited the sleep clinic of our hospital (General Hospital of Thessaloniki “G. Papanikolaou”) between September 2010 and September 2023 were reviewed retrospectively. Those considered eligible for inclusion in this study were those who were current or former smokers (adults who have smoked 100 cigarettes in their lifetime and who currently smoke cigarettes, or had quit smoking at the time of interview, respectively) [23] and consented to participate in this study. Patients were excluded if there were no data about their smoking history (packyears of smoking and Fagerström scale) [24,25]. As a result, a total of 2359 patients were included in the analysis. Some of the patients did not answer all the questions of each questionnaire. The Fagerström scale in particular was answered only by those participants who were current smokers at the time of their visit.

Data collection: All the data that were used in the present study originated, either from the questionnaires that the patients were asked to answer during their visit, or by the history obtaining and the physical examination that was performed during the patients’ visit by an experienced respiratory physician with specialization in sleep medicine.

### 2.3. Anthropometric Characteristics

The participants’ baseline characteristics such as their age, sex, body mass index (BMI), neck, waist and hip circumference, Malampati score, blood pressure, heart rate, SaO_2_ and family status were recorded.

### 2.4. Comorbidities

Participants also provided information about their alcohol consumption and cardiovascular, respiratory, metabolic, and psychiatric comorbidities.

### 2.5. Sleep Quality Characteristics

Additionally, they also answered questionnaires regarding their sleep latency, night sleep and nap duration, sleep disturbances such as nightmares, restless sleep, abnormal movements, sleeptalking and legs’ movements, the Berlin and STOP bang questionnaires [26,27], the Epworth Sleepiness Scale (ESS) [28], the Rosenberg self-esteem scale [29] and the Athens insomnia scale (AIS) [30].

### 2.6. OSA-Related Symptoms

They also provided answers to questions related to OSA symptoms, such as bad mood, heavy head, headaches, morning fatigue, memory loss, dry mouth, choking or breathing pauses during sleep, night awakenings, snoring loudness and frequency, dropping thing from hands due to sleepiness and needing a passenger when driving to be kept awake.

### 2.7. Polysomnography

The participants were also subjected to either type 1 or type 3 sleep studies [31], which were scored according to the guidelines that were applicable at any given time period [32]. Due to limited resources, the majority of our patients (86.4%) were subjected to type 3 sleep study, while type 1 sleep study (13.6%) was applied in patients who, by their history or physical examination, caused high suspicion of suffering from additional sleep disorders, other than OSAHS. A diagnosis of OSA existence and severity was made based on apnea-hypopnea index (AHI), (OSA diagnosis: AHI ≥ 5, mild OSA: 5 ≥ AHI > 15, moderate OSA: 15 ≥ AHI > 30, severe OSA: AHI ≥ 30) [33,34].

### 2.8. Statistical Analysis

Statistical analysis was performed using the SPSS (version 20 IBM SPSS statistical software, Armonk, NY, USA). Continuous variables are presented as mean ± SD. Statistical significance was accepted at *p* < 0.05. Kolmogorov–Smirnov test was used to separate parametric from non-parametric variables. To detect significant differences in packyears of smoking and in Fagerström scale between categorical variables, the independent samples *t*-test or the one-way ANOVA test were used for parametric variables and the Mann–Whitney U test or the Kruskal–Wallis test were used for non-parametric variables, depending on the categorical variable being dichotomous or not. In the latter, a post hoc analysis using the Bonferroni test for parametric variables or the Mann–Whitney U test for non-parametric variables was also performed. Since some of the patients did not answer all the questions of each questionnaire, the total number of the patients that was included in each statistical test is given in the tables of this study. To detect significant correlations between the packyears of smoking or the Fagerström scale and the other continuous variables of this study, linear regression was used.

## 3. Results

### 3.1. Anthropometric Characteristics

Males had a history of smoking higher than females (39.7 ± 33.0 vs. 26.0 ± 23.0, *p* < 0.001), while the same applied for married and divorced smokers compared to singles and widowers (38.2 ± 31.4 and 39.2 ± 37.6 vs. 23.5 ± 23.4 and 16.0 ± 13.2, respectively, *p* < 0.001). Age, BMI, and neck and waist circumference correlated significantly with both packyears and the Fagerström scale; however, the correlation with packyears was stronger compared to that with the Fagerström scale (r = 0.14–0.39, *p* < 0.001 vs. r = 0.08–0.12, *p* = 0.007–0.025, respectively). SaO_2_ showed a significant negative correlation with both packyears and the Fagerström scale (r = −0.19, *p* < 0.001 and r = −0.12, *p* = 0.006, respectively), while heart rate showed a negative correlation with packyears (r = −0.07, *p* = 0.002) and a positive one with the Fagerström scale (r = 0.10, *p* = 0.023) (Table 1).

### 3.2. Comorbidities

As far as comorbidities, smoking history in packyears was significantly higher in patients with hypertension (44.4 ± 35.4 vs. 31.9 ± 27.8, *p* < 0.001), diabetes mellitus (50.0 ± 38.6 vs. 34.1 ± 29.3, *p* < 0.001), coronary disease (54.0 ± 42.4 vs. 34.3 ± 28.9, *p* < 0.001), acute myocardial infarction (52.9 ± 42.6 vs. 36.2 ± 31.0, *p* = 0.001), heart failure (64.3 ± 51.9 vs. 36.6 ± 31.3, *p* = 0.032), hyperlipidemia (44.7 ± 34.8 vs. 35.0 ± 30.6, *p* < 0.001), ischemic stroke (43.9 ± 34.0 vs. 36.6 ± 31.6, *p* = 0.049) and pulmonary disease (55.2 ± 40.6 vs. 34.8 ± 29.9, *p* < 0.001), while it was significantly lower in patients with hypothyroidism (32.6 ± 29.9 vs. 37.2 ± 31.8, *p* = 0.034). On the other hand, the Fagerström scale was significantly higher only in patients with hypertension (5.4 ± 2.7 vs. 4.7 ± 2.8, *p* = 0.002), diabetes mellitus (5.5 ± 2.9 vs. 4.9 ± 2.8, *p* = 0.032) and hyperlipidemia (5.8 ± 2.4 vs. 4.8 ± 2.8, *p* < 0.001). Furthermore, patients who were consuming alcohol every day exhibited significantly higher smoking history in packyears compared to those who were drinking alcohol less frequently (*p* = 0.001), while the same did not apply for the Fagerström scale (Table 2).

### 3.3. Sleep Quality Characteristics

Both packyears of smoking and the Fagerström scale were negatively correlated with night sleep duration and the Rosenberg self-esteem scale (r = −0.10, *p* < 0.001 and r = −0.06, *p* = 0.036, respectively, for packyears of smoking and r = −0.12, *p* = 0.001 and r = −0.13, *p* < 0.001, respectively, for the Fagerström scale) and positively correlated with AIS (r = 0.06, *p* = 0.012 and r = 0.21, *p* < 0.001, respectively), while packyears of smoking were also positively correlated with ESS (r = 0.06, *p* = 0.007). Patients at high risk for sleep apnea in the Berlin questionnaire exhibited a history of smoking with a higher number of packyears and a nicotine dependence with a higher Fagerström scale compared to those at low risk (38.0 ± 31.7 vs. 28.0 ± 29.0, *p* < 0.001 and 5.1 ± 2.8 vs. 4.1 ± 2.6, *p* < 0.001, respectively). The same also applied for the STOP bang questionnaire (37.4 ± 31.7 vs. 15.3 ± 11.9, *p* < 0.001 and 5.1 ± 2.8 vs. 3.4 ± 2.6, *p* = 0.001, respectively). Patients with abnormal movements during sleep and those with restless sleep showed a significantly higher nicotine dependence, measured with the Fagerström scale, compared to those without abnormal movements or restless sleep (5.4 ± 2.8 vs. 4.7 ± 2.8, *p* = 0.002 and 5.1 ± 2.9 vs. 4.7 ± 2.7, *p* = 0.043), something that was not the case for packyears of smoking; however, increasing frequency of leg movements during sleep was linked with a significantly higher smoking history, measured in packyears (*p* < 0.001), something that was not evident for nicotine dependence (Table 3).

### 3.4. OSA-Related Symptoms

As far as sleep-related symptoms, patients who drop things from their hands due to sleepiness or need a passenger when driving in order to be kept awake showed higher nicotine dependence (5.4 ± 3.0 vs. 4.8 ± 2.7, *p* = 0.024 and 5.6 ± 3.0 vs. 4.8 ± 2.7, *p* = 0.003, respectively). They also exhibited an increasing nicotine dependence with increasing snoring loudness and breathing pauses frequency (*p* = 0.001 and *p* < 0.001, respectively). Smoking history, measured with packyears of smoking, also exhibited an increasing trend with increasing frequency of breathing pauses and night awakenings (*p* = 0.011 and *p* < 0.001, respectively) (Table 4).

### 3.5. Polysomnography

As far as sleep study parameters, packyears positively correlated with %REM sleep time (r = 0.19, *p* = 0.042), AHI (r = 0.10, *p* < 0.001), central apneas (r = 0.06, *p* = 0.002), oxygen desaturation index (ODI) (r = 0.10, *p* < 0.001), mean and maximum apnea duration (r = 0.10, *p* < 0.001 and r = 0.11, *p* < 0.001, respectively), while they were negatively correlated with % non-REM sleep time (r = −0.19, *p* = 0.042) and mean and minimum SaO_2_ (r = −0.18, *p* < 0.001 and r = −0.13, *p* < 0.001, respectively). Furthermore, smoking history exhibited a significantly increasing trend with increasing OSA diagnosis and severity (*p* < 0.001). On the other hand, the Fagerström scale showed a significant positive correlation only with AHI and ODI (r = 0.09, *p* = 0.016 and r = 0.08, *p* = 0.025, respectively) and a significant negative correlation with total sleep time (r = −0.40, *p* = 0.021) and mean and minimum SaO_2_ (r = −0.17, *p* < 0.001 and r = −0.10, *p* = 0.004, respectively), while its increasing trend with increasing OSA diagnosis and severity was not as evident as that of smoking history in packyears (*p* = 0.003) (Table 5).

## 4. Discussion

### 4.1. Main Findings

There are several noteworthy results in this cross-sectional study. The main finding is that smoking history in packyears seems to be a better quantitative index than nicotine dependence, measured with the Fagerström scale, in the picturing of the relationship between smoking and OSAHS presentation and severity, whereas, the Fagerström scale seems to excel in the quantification of the relationship of smoking with other abnormal sleep behaviors.

### 4.2. Anthropometric Characteristics and Comorbidities

Smoking history in packyears, rather than nicotine dependence, was strongly correlated with some epidemiological characteristics such as age, BMI and neck and waist circumference; in turn, patients with a heavier smoking history, who are also older and more obese, suffer more from metabolic, cardiovascular and respiratory comorbidities. It is well established that age and obesity deteriorate OSAHS [35], and subsequently, severe OSAHS is responsible for increasing the prevalence of metabolic and cardiovascular comorbidities [36]. Furthermore, it has been shown that smoking history may increase OSA severity, both in the present and in previous studies [37,38,39,40]. Consequently, smoking might contribute to the emergence of metabolic and cardiovascular comorbidities in OSA patients, both directly, by dysregulating metabolic pathways and damaging vascular endothelium and indirectly, by contributing to obesity and deteriorating OSA.

OSA-related symptoms and polysomnography parameters: Regarding sleep study parameters, the Fagerström scale exhibited a positive correlation with AHI and ODI and a negative correlation with total sleep time and mean and minimum SaO_2_; however, those correlations were more significant with packyears of smoking, which also showed a significant positive correlation with %REM sleep time, central apneas and mean and maximum apnea duration. Our findings about the positive correlation between smoking history in packyears and %REM sleep time are in accordance with other studies, which have shown a similar relationship between smoking and REM sleep [41,42]. The fact that in our study correlation was found only for packyears of smoking and not for the Fagerström scale possibly suggests that this finding concerns the whole population of smokers (ex- and current), while in current smokers, who have answered the Fagerström scale, this relationship was not evident. The rest of the correlations, either positive or negative, that were found in this research between the Fagerström scale and packyears of smoking, on the one hand, and AHI, ODI, total sleep time, mean and minimum SaO_2_, mean and maximum apnea duration, on the other hand, are in accordance with the findings of previous studies [8,43,44,45,46,47,48,49,50,51,52]. Furthermore, packyears of smoking were positively correlated with ESS, probably because they are related with higher AHI and more severe OSA [8,44,48,50]. In contrast with sleep study parameters, sleep-related symptoms such as dropping things from hands due to sleepiness, needing a passenger during driving in order to be kept awake, snoring loudness and breathing pauses frequency seem to be related more with the Fagerström scale, rather than with packyears of smoking, which are more related only with the frequency of breathing pauses and night awakenings.

### 4.3. Sleep Quality Characteristics

Apart from OSAHS-related results, this research also found relationships between smoking indices and other sleep parameters. Both packyears and the Fagerström scale were positively correlated with AIS, meaning that smoking history and nicotine dependence are linked with insomnia, something that was also shown in previous studies [53,54,55,56]. Furthermore, restless legs frequency was also linked with smoking history, as in previous studies [42,57]; however, patients with abnormal movements during sleep and those with restless sleep showed a significantly higher nicotine dependence, measured with the Fagerström scale, compared to those without abnormal movements or restless sleep, something that was not the case for packyears of smoking, meaning that those abnormal behaviors during sleep are related more with active smoking, rather than with smoking history. Previous studies have exhibited the relationship between smoking and abnormal sleep behaviors, such as parasomnias [16]. The better alignment of the Fagerström scale instead of packyears of smoking with abnormal sleep behaviors might be connected with the fact that current smokers present a smoking-dependent negative association with REM sleep, along with lower delta power and higher alpha power in electroencephalogram, and increased wake time after sleep onset with diminished sleep continuity, compared to former or never smokers [41,58]. Furthermore, current smokers also experience a dose-dependent decrease in the N3 sleep stage with a respective increase in N1 and N2 stages [14,43,45,49], but increased REM sleep density due to their longer sleep latency and shorter sleep time in general [42].

### 4.4. Limitations

There are several limitations in this study; the main one is its cross-sectional nature, which forbids the establishment of a causative relationship between smoking and sleep, since the temporal sequence between them cannot be determined in cross-sectional studies. Another limitation is that the majority of the participants were subjected to a type 3 sleep study (86.4%), instead of a type 1 (13.6%). Type 3 sleep studies are less precise compared to type 1, as they usually underestimate AHI, something that constitutes a bias. Furthermore, the reliance on type 3 studies prevents from evaluating sleep stages and behaviors in a more detailed manner. Consequently, the majority of the study’s data were not recorded with objective means, but were reported by the patients or their bed partners mainly through questionnaires, meaning that they are subjective and introduce a potential recall bias. Another important limitation is the fact that during the 13-year period of this study, some patients did not consent to the use of their data for research purposes, or omitted to answer all the questions of each questionnaire, a fact that constitutes a participants’ bias. Furthermore, some of the questions that were designed by us, regarding mainly the frequency of the sleep-related symptoms, were not previously validated, something that might constitute another possible bias of this study.

## 5. Conclusions

Despite these limitations, this study included a rather large number of patients, compared to other studies in this field, and showed a relationship between nicotine use and dependence and OSAHS presentation and severity, with this relationship being stronger for smoking history than for nicotine dependence. Furthermore, nicotine dependence seemed to be related more with abnormal behaviors during sleep. Since there are several studies which suggest a deleterious relationship between smoking and sleep, there is a need for cohort studies that could establish a temporal sequence between those two, something that will allow more targeted therapeutic approaches.

## Figures and Tables

**Table 1 healthcare-13-00049-t001:** Correlations and comparisons between anthropometric characteristics and packyears of smoking and Fagerström scale.

Characteristic		Packyears of Smoking			Fagerström Scale		
Comparisons		Mean ± Standard Deviation		*p* (Value)	Mean ± Standard Deviation		*p* (Value)
Gender	Female	26.0 ± 23.0 (N = 505)		<0.001	4.8 ± 2.7 (N = 179)		0.45
	Male	39.7 ± 33.0 (N = 1854)			5.0 ± 2.8 (N = 610)		
Family status	Single	23.5 ± 23.4 (N = 312)		<0.001	4.5 ± 2.9 (N = 149)		0.28
	Married	38.2 ± 31.4 (N = 1552)			5.1 ± 2.8 (N = 514)		
	Divorced	39.2 ± 37.6 (N = 75)			5.0 ± 2.8 (N = 36)		
	Widower	16.0 ± 13.2 (N = 3)			4.0 ± 0.0 (N = 1)		
Correlations		B (95% C.I.)	r	*p* (Value)	B (95% C.I.)	r	*p* (Value)
Age (years)		0.98 (0.88–1.07)	0.39	<0.001	0.02 (0.00–0.04)	0.09	0.016
BMI (Kg/m^2^)		0.61 (0.43–0.79)	0.14	<0.001	0.03 (0.00–0.06)	0.08	0.023
Neck circumference (cm)		0.84 (0.61–1.08)	0.17	<0.001	0.05 (0.01–0.08)	0.10	0.025
Waist circumference (cm)		0.41 (0.32–0.50)	0.23	<0.001	0.02 (0.01–0.03)	0.12	0.007
Hip circumference (cm)		0.26 (0.15–0.37)	0.13	<0.001	0.01 (−0.00–0.03)	0.06	0.23
Malampati score		7.01 (−1.83–15.86)	0.12	0.12	0.54 (−0.47–1.55)	0.14	0.29
SaO_2_ (%)		−2.42 (−3.02–−1.82)	−0.19	<0.001	−0.19 (−0.32–−0.06)	−0.12	0.006
Heart rate (beats/min)		−0.18 (−0.30–−0.06)	−0.07	0.002	0.02 (0.00–0.04)	0.10	0.023
Systolic blood pressure (mmHg)		0.08 (−0.11–0.27)	0.04	0.39	0.02 (−0.00–0.05)	0.13	0.08
Diastolic blood pressure (mmHg)		−0.11 (−0.41–0.20)	−0.03	0.50	0.04 (−0.00–0.08)	0.15	0.05

N = number, C.I. = confidence intervals, m = meters, Kg = kilograms, cm = centimeters, mmHg = millimeters of Mercury.

**Table 2 healthcare-13-00049-t002:** Comparisons of packyears of smoking and Fagerström scale between patients with and without comorbidities.

Comorbidity			Packyears of Smoking		Fagerström Scale	
			Mean ± Standard Deviation	*p* (Value)	Mean ± Standard Deviation	*p* (Value)
Alcohol	Almost never		37.5 ± 33.0 (N = 675)	0.001	5.0 ± 2.7 (N = 224)	0.21
	A few times per month		36.0 ± 29.9 (N = 1284)		4.9 ± 2.8 (N = 401)	
	1–2 times per week		34.0 ± 29.9 (N = 166)		4.9 ± 2.8 (N = 73)	
	3–5 times per week		35.5 ± 28.8 (N = 103)		4.7 ± 2.8 (N = 50)	
	Every day		50.8 ± 47.7 (N = 86)		6.0 ± 3.0 (N = 28)	
Hypertension		Yes	44.4 ± 35.4 (N = 928)	<0.001	5.4 ± 2.7 (N = 232)	0.002
		No	31.9 ± 27.8 (N = 1431)		4.7 ± 2.8 (N = 557)	
Diabetes Mellitus		Yes	50.0 ± 38.6 (N = 398)	<0.001	5.5 ± 2.9 (N = 92)	0.032
		No	34.1 ± 29.3 (N = 1961)		4.9 ± 2.8 (N = 697)	
Coronary disease		Yes	54.0 ± 42.4 (N = 302)	<0.001	5.4 ± 2.9 (N = 60)	0.23
		No	34.3 ± 28.9 (N = 2057)		4.9 ± 2.8 (N = 729)	
Acute myocardial infarction		Yes	52.9 ± 42.6 (N = 83)	0.001	4.7 ± 3.3 (N = 16)	0.72
		No	36.2 ± 31.0 (N = 2276)		4.9 ± 2.8 (N = 773)	
Heart failure		Yes	64.3 ± 51.9 (N = 19)	0.032	1.0 ± 0.0 (N = 1)	0.16
		No	36.6 ± 31.3 (N = 2340)		4.9 ± 2.8 (N = 788)	
Arrythmia		Yes	38.0 ± 30.8 (N = 288)	0.48	4.6 ± 2.8 (N = 68)	0.30
		No	36.6 ± 31.8 (N = 2071)		5.0 ± 2.8 (N = 721)	
Hyperlipidemia		Yes	44.7 ± 34.8 (N = 433)	<0.001	5.8 ± 2.4 (N = 93)	<0.001
		No	35.0 ± 30.6 (N = 1925)		4.8 ± 2.8 (N = 696)	
Ischemic stroke		Yes	43.9 ± 34.0 (N = 74)	0.049	4.9 ± 2.2 (N = 19)	0.95
		No	36.6 ± 31.6 (N = 2282)		4.9 ± 2.8 (N = 768)	
Pulmonary disease		Yes	55.2 ± 40.6 (N = 228)	<0.001	5.2 ± 2.8 (N = 48)	0.56
		No	34.8 ± 29.9 (N = 2131)		4.9 ± 2.8 (N = 741)	
Hypothyroidism		Yes	32.6 ± 29.9 (N = 236)	0.034	4.6 ± 2.9 (N = 67)	0.37
		No	37.2 ± 31.8 (N = 2123)		5.0 ± 2.8 (N = 722)	
Depression		Yes	37.6 ± 32.4 (N = 70)	0.83	5.9 ± 2.4 (N = 29)	0.05
		No	36.8 ± 31.6 (N = 2289)		4.9 ± 2.8 (N = 760)	

**Table 3 healthcare-13-00049-t003:** Correlations and comparisons between sleep quality characteristics and packyears of smoking and Fagerström scale.

Sleep Quality Characteristic		Packyears of Smoking			Fagerström Scale		
Correlations		B (95% C.I.)	r	*p* (Value)	B (95% C.I.)	r	*p* (Value)
Night sleep duration (h)		−4.11 (−5.74–−2.48)	−0.10	<0.001	−0.46 (−0.73–−0.19)	−0.12	0.001
Sleep latency (min)		0.78 (−0.30–1.87)	0.03	0.16	0.16 (−0.01–0.34)	0.07	0.06
Nap duration (h)		0.29 (−2.64–3.22)	0.01	0.85	0.32 (−0.17–0.81)	0.06	0.20
Epworth sleepiness scale		0.38 (0.10–0.65)	0.06	0.007	0.03 (−0.01–0.07)	0.05	0.14
Athens insomnia scale		0.31 (0.07–0.55)	0.06	0.012	0.11 (0.08–0.15)	0.21	<0.001
Rosenberg self-esteem scale		−0.33 (−0.63–−0.02)	−0.06	0.036	−0.08 (−0.12–−0.04)	−0.13	<0.001
Comparisons		Mean ± standard deviation		*p* (Value)	Mean ± standard deviation		*p* (Value)
Berlin questionnaire	Low risk	28.0 ± 29.0 (N = 295)		<0.001	4.1 ± 2.6 (N = 120)		<0.001
	High risk	38.0 ± 31.7 (N = 2022)			5.1 ± 2.8 (N = 664)		
STOP bang questionnaire	Low risk	15.3 ± 11.9 (N = 62)		<0.001	3.4 ± 2.6 (N = 508)		0.001
	High risk	37.4 ± 31.7 (N = 1555)			5.1 ± 2.8 (N = 33)		
Nightmares	Yes	35.9 ± 29.6 (N = 637)		0.42	5.1 ± 2.8 (N = 212)		0.35
	No	37.1 ± 32.4 (N = 1722)			4.9 ± 2.8 (N = 577)		
Sleeptalking	Yes	37.1 ± 30.5 (N = 734)		0.73	5.2 ± 2.9 (N = 260)		0.13
	No	36.6 ± 32.2 (N = 1625)			4.8 ± 2.8 (N = 529)		
Abnormal movements during sleep	Yes	38.3 ± 32.1 (N = 754)		0.10	5.4 ± 2.8 (N = 259)		0.002
	No	36.1 ± 31.4 (N = 1605)			4.7 ± 2.8 (N = 530)		
Messing the bed during sleep/Restless sleep	Yes	37.2 ± 31.4 (N = 1387)		0.47	5.1 ± 2.9 (N = 451)		0.043
	No	36.2 ± 32.0 (N = 972)			4.7 ± 2.7 (N = 338)		
Legs’ movements	Don’t know	32.1 ± 30.9 (N = 96)		<0.001	4.8 ± 3.1 (N = 35)		0.06
	Never	33.2 ± 27.1 (N = 563)			4.7 ± 2.7 (N = 232)		
	Rarely	29.6 ± 25.8 (N = 161)			4.3 ± 2.8 (N = 77)		
	Sometimes	39.3 ± 32.5 (N = 612)			5.3 ± 2.6 (N = 158)		
	Usually	38.8 ± 34.0 (N = 740)			5.1 ± 2.8 (N = 226)		
	Always	40.1 ± 34.8 (N = 163)			5.2 ± 3.0 (N = 60)		

**Table 4 healthcare-13-00049-t004:** Comparison of packyears of smoking and Fagerström scale in patients with obstructive sleep apnea-related symptoms.

Obstructive Sleep Apnea-Related Symptom		Packyears of Smoking		Fagerström Scale	
		Mean ± Standard Deviation	*p* (Value)	Mean ± Standard Deviation	*p* (Value)
Dry mouth	Don’t Know	30.8 ± 17.6 (N = 6)	0.40	5.0 ± 0.0 (N = 1)	0.08
	Almost never	35.3 ± 30.5 (N = 667)		4.9 ± 2.8 (N = 249)	
	1–2 times per month	47.7 ± 37.0 (N = 16)		4.3 ± 3.8 (N = 4)	
	1–2 times per week	41.7 ± 43.4 (N = 48)		4.7 ± 3.0 (N = 11)	
	3–4 times per week	36.1 ± 38.5 (N = 55)		2.0 ± 2.2 (N = 8)	
	Daily	37.2 ± 31.4 (N = 1550)		5.0 ± 2.8 (N = 514)	
Morning fatigue	Don’t Know	26.9 ± 23.0 (N = 4)	0.23	5.5 ± 2.1 (N = 2)	0.044
	Almost never	36.2 ± 30.4 (N = 694)		4.5 ± 2.5 (N = 230)	
	1–2 times per month	40.7 ± 42.8 (N = 32)		5.3 ± 3.4 (N = 8)	
	1–2 times per week	43.8 ± 36.8 (N = 57)		4.8 ± 2.8 (N = 11)	
	3–4 times per week	31.5 ± 25.3 (N = 96)		3.8 ± 2.7 (N = 19)	
	Daily	37.0 ± 32.0 (N = 1460)		5.2 ± 2.9 (N = 518)	
Bad mood	Almost never	35.8 ± 30.4 (N = 695)	0.52	4.5 ± 2.4 (N = 230)	0.024
	1–2 times per month	39.4 ± 30.3 (N = 32)		5.3 ± 3.2 (N = 9)	
	1–2 times per week	41.9 ± 38.4 (N = 60)		3.9 ± 3.1 (N = 15)	
	3–4 times per week	34.4 ± 29.2 (N = 104)		4.7 ± 2.8 (N = 21)	
	Almost daily	37.1 ± 32.0 (N = 1450)		5.2 ± 2.9 (N = 513)	
	Daily	15.0 ± 4.2 (N = 2)		±(N = 0)	
Headache	Don’t Know	12.0 ± 0.0 (N = 1)	0.49	±(N = 0)	0.36
	Almost never	37.1 ± 32.4 (N = 1489)		4.8 ± 2.7 (N = 550)	
	1–2 times per month	30.9 ± 24.0 (N = 72)		4.5 ± 3.1 (N = 15)	
	1–2 times per week	36.6 ± 26.6 (N = 135)		4.9 ± 2.4 (N = 22)	
	3–4 times per week	33.9 ± 27.1 (N = 122)		5.2 ± 2.8 (N = 35)	
	Daily	37.4 ± 32.2 (N = 523)		5.3 ± 3.1 (N = 166)	
Heavy head	Don’t Know	15.7 ± 14.8 (N = 3)	0.34	± (N = 0)	0.37
	Almost never	37.0 ± 32.4 (N = 1479)		4.8 ± 2.7 (N = 546)	
	1–2 times per month	30.6 ± 22.9 (N = 63)		4.4 ± 2.9 (N = 13)	
	1–2 times per week	34.3 ± 24.5 (N = 115)		5.2 ± 2.2 (N = 17)	
	3–4 times per week	35.0 ± 29.5 (N = 128)		5.0 ± 2.8 (N = 37)	
	Daily	37.9 ± 31.9 (N = 555)		5.3 ± 3.1 (N = 175)	
Memory loss	Yes	37.5 ± 32.1 (N = 1174)	0.27	5.1 ± 2.8 (N = 364)	0.12
	No	36.1 ± 31.2 (N = 1185)		4.8 ± 2.8 (N = 425)	
Drops things from hands	Yes	39.2 ± 34.2 (N = 527)	0.06	5.4 ± 3.0 (N = 163)	0.024
	No	36.1 ± 30.9 (N = 1832)		4.8 ± 2.7 (N = 626)	
Need a passenger when driving to be kept awake	Yes	39.2 ± 29.8 (N = 478)	0.06	5.6 ± 3.0 (N = 146)	0.003
	No	36.2 ± 32.1 (N = 1881)		4.8 ± 2.7 (N = 643)	
Snoring frequency	Never	41.8 ± 37.4 (N = 18)	0.39	±(N = 0)	0.08
	Almost never	38.9 ± 42.0 (N = 54)		4.1 ± 2.7 (N = 19)	
	1–2 times per month	62.2 ± 87.5 (N = 6)		2.5 ± 0.7 (N = 2)	
	1–2 times per week	24.4 ± 18.6 (N = 8)		5.0 ± 2.8 (N = 2)	
	3–4 times per week	43.2 ± 70.0 (N = 11)		1.0 ± 1.0 (N = 3)	
	Every night	36.6 ± 30.7 (N = 2221)		5.0 ± 2.8 (N = 751)	
	Many times per night	36.6 ± 35.0 (N = 41)		4.4 ± 3.2 (N = 12)	
Snoring loudness	Slightly loud	35.3 ± 38.2 (N = 47)	0.30	3.9 ± 2.8 (N = 16)	0.001
	A little loud	36.5 ± 45.2 (N = 59)		4.0 ± 2.7 (N = 17)	
	Mediocrely loud	33.9 ± 31.6 (N = 186)		3.9 ± 2.8 (N = 69)	
	Very loud	36.5 ± 30.4 (N = 1721)		5.0 ± 2.8 (N = 568)	
	Extremely loud	39.9 ± 33.6 (N = 326)		5.4 ± 2.8 (N = 119)	
Breathing pauses during sleep	Almost never	32.4 ± 34.0 (N = 208)	0.011	3.9 ± 2.6 (N = 95)	<0.001
	1–2 times per month	29.0 ± 28.9 (N = 8)		4.3 ± 2.9 (N = 3)	
	1–2 times per week	22.2 ± 18.5 (N = 24)		2.2 ± 3.0 (N = 5)	
	3–4 times per week	25.9 ± 20.5 (N = 33)		5.5 ± 2.4 (N = 13)	
	Every night	37.6 ± 31.4 (N = 1784)		5.1 ± 2.8 (N = 559)	
	Many times per night	36.7 ± 31.5 (N = 279)		4.9 ± 2.8 (N = 112)	
Night awakenings	Almost never	30.7 ± 27.2 (N = 534)	<0.001	4.9 ± 2.6 (N = 240)	0.32
	1–2 times per month	31.2 ± 25.7 (N = 29)		5.4 ± 3.2 (N = 12)	
	1–2 times per week	28.3 ± 25.2 (N = 58)		4.1 ± 2.9 (N = 34)	
	3–4 times per week	37.9 ± 31.5 (N = 1596)		5.0 ± 2.8 (N = 485)	
	Every night	54.6 ± 44.2 (N = 123)		5.7 ± 3.1 (N = 17)	

**Table 5 healthcare-13-00049-t005:** Correlations and comparisons between polysomnography parameters and packyears of smoking and Fagerström scale.

Polysomnography Parameter		Packyears of Smoking			Fagerström Scale		
Correlations		B (95% C.I.)	r	*p* (Value)	B (95% C.I.)	r	*p* (Value)
Total sleep time (min)		−0.03 (−0.12–0.05)	−0.07	0.45	−0.01 (−0.02–−0.00)	−0.40	0.021
% REM sleep time (%)		0.60 (0.02–1.17)	0.19	0.042	−0.06 (−0.17–0.04)	−0.23	0.21
% Non-REM sleep time (%)		−0.60 (−1.17–−0.02)	−0.19	0.042	0.06 (−0.04–0.17)	0.23	0.21
AHI (events/h)		0.13 (0.08–0.18)	0.10	<0.001	0.01 (0.00–0.02)	0.09	0.016
Central apneas (events/h)		0.90 (0.33–1.47)	0.06	0.002	−0.02 (−0.14–0.09)	−0.01	0.71
Mean SaO_2_ (%)		−1.70 (−2.07–−1.33)	−0.18	<0.001	−0.15 (−0.20–−0.09)	−0.17	<0.001
Minimum SaO_2_ (%)		−0.42 (−0.55–−0.29)	−0.13	<0.001	−0.03 (−0.05–−0.01)	−0.10	0.004
ODI (events/h)		0.13 (0.08–0.18)	0.10	<0.001	0.01 (0.00–0.02)	0.08	0.025
Mean apnea duration (s)		0.42 (0.23–0.61)	0.10	<0.001	0.01 (−0.02–0.04)	0.03	0.39
Maximum apnea duration (s)		0.14 (0.08–0.20)	0.11	<0.001	0.01 (−0.01–0.02)	0.04	0.33
Comparisons		Mean ± standard deviation		*p* (Value)	Mean ± standard deviation		*p* (Value)
OSA diagnosis	Absent (AHI < 5)	25.9 ± 31.8 (N = 259)		<0.001	4.2 ± 2.9 (N = 128)		0.003
	Mild (AHI 5–15)	31.5 ± 26.7 (N = 290)			5.0 ± 2.7 (N = 118)		
	Moderate (AHI 15–30)	36.7 ± 31.0 (N = 611)			4.8 ± 2.7 (N = 200)		
	Severe (AHI > 30)	40.5 ± 32.4 (N = 1199)			5.3 ± 2.8 (N = 343)		

N = number, REM = rapid eye movement, AHI = apnea-hypopnea index, ODI = oxygen desaturation index, OSA = obstructive sleep apnea-hypopnea syndrome.

## Data Availability

The data that support the findings of this study are available from the corresponding author upon reasonable request.

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
