# Peer review of "Smoking History and Nicotine Dependence Alter Sleep Features in Patients with Obstructive Sleep Apnea-Hypopnea Syndrome"

_healthcare, 2024, doi:10.3390/healthcare13010049_

Round 1

Reviewer 1 Report

Comments and Suggestions for Authors

I am grateful for the opportunity to review the article Nicotine dependence alters disease characteristics in patients with obstructive sleep apnea -hypopnea syndrome submitted to the Healthcare Journal MDPI.

The article has the correct structure and is generally written in the correct language.

However, I consider some points essential to be clarified by the authors so that, in my opinion, the article can be accepted for publication.

1) This type of study, with this objective, is not inspiring. Did the authors consider including words in a Greek population as part of their objective?

2) The Methods should identify the Univeristy of Thessaloniki, Greece.

3) When reading the study methodology, the authors imply that they carried out the entire study, collection, examinations, etc... Did the authors do this over 13 years or use the data in the patients' histories? If so, they should clarify.

4) Where were the patients seen? I assume it is in a hospital, but it has not been identified.

5) Were all patients submitted to all examinations and questionnaires?

6) Sleep studies are excluded from the previous question. What were the criteria for applying type 1 or type 3?

7) The results present several problems. Tables should not be "broken" between pages. I suggest increasing or formatting the tables so this doesn't happen.

8) With the sample being 2359 patients – in Table 1, we have a total for Family Status of only 1942 and 700, respectively. What happened to the remaining patients? If they do not have information, they must include an item without information in the table.9) In Table 2 – Do the authors not include the item never in alchohol? The total number of patients does not coincide with the total sample – 2314 and 776.

10) Table 3 shows missing patients from the sample in the Berlin questionnaire (42 and 5 patients are missing in each column) and in the Stop Bang questionnaire. What happened? If you don't have the data, you must declare it.

11) Regarding Legs movements, there are also patients missing in total from both columns (24 in column 1 and 1 in column 2). What happened?

12) Table 4 – Dry Mouth patients are missing in both columns: Morning fatigue, heavy Headache, and Headache.

13) In Bad Mood, in Table 4, sample members are missing. Do they all have a bad mood? There is no answer category "never". The same applies to Snoring loudness, breathing pauses during sleep, and night awakenings. Is there a lack of patients/responses, and has no one ever responded?

14) The discussion seems short to me, and the limitations should not be in the conclusion but in the discussion.

Thank you.

Author Response

Comments 1: This type of study, with this objective, is not inspiring. Did the authors consider including words in a Greek population as part of their objective?

Response 1: Thank you for pointing that out. During the study period, our sleep clinic was one of the only three sleep clinics of the public sector in Northern Greece and the only one that was operating a smoking cessation clinic simultaneously, covering a population of almost 2 million people. During the study period our sleep and smoking cessation clinics were publicized by the local media, as part of a campaign for the promotion of public health, and as a result, more than 10.000 people visited our sleep clinic during this period, therefore recruitment was not a problem in our study, however, there were many patients that visited our sleep clinic during the study period, who, despite being current or ex-smokers, had not provided consent, or information about their nicotine dependence or smoking history, something that could cause a participant’s bias in our study. This is something that we mention in the revised version of our manuscript.

Comments 2: The Methods should identify the Univeristy of Thessaloniki, Greece.

Response 2: Thank you for pointing that out. In our revised manuscript, we have changed this part accordingly.

Comments 3: When reading the study methodology, the authors imply that they carried out the entire study, collection, examinations, etc... Did the authors do this over 13 years or use the data in the patients' histories? If so, they should clarify.

Response 3: Thank you for pointing that out. We have clarified in the “Introduction” section of our revised manuscript, that the records of all adult patients who have visited the sleep clinic of our hospital between September 2010 and September 2023 were reviewed retrospectively, and eligible for inclusion in the study were considered those who were current or former smokers and consented to participate in the study. When our patients visit our sleep clinic, they are given a pack of papers with different questions and questionnaires to answer, i.e. the Berlin and STOP bang questionnaires, the Epworth Sleepiness Scale (ESS), the Rosenberg self-esteem scale, the Athens insomnia scale (AIS), the Fagerström scale etc. At the end of this pack of papers, there is a question regarding the patients’ consent for the use of all this information for research purposes. This consent might concern different studies, even studies that have not been designed at the time of consent, a fact that is explained to the patients before they give their consent. We only use the information of the patients that provide this consent and only after obtaining the respective approval by our hospital’s or our university’s ethics committees, which are always aware of the procedure by which we obtained this consent.

Comments 4: Where were the patients seen? I assume it is in a hospital, but it has not been identified.

Response 4: Thank you for pointing that out. In the “Introduction” section of our revised manuscript we identify our hospital as the General hospital of Thessaloniki “G. Papanikolaou”.

Comments 5: Were all patients submitted to all examinations and questionnaires?

Response 5: Thank you for your comment. Yes, all patients were submitted to all examinations and questionnaires, however, some of them did not answer all the questions of each questionnaire. This might have caused a participants’ bias, a fact that we mention in the “limitations” section of our revised manuscript.

Comments 6: Sleep studies are excluded from the previous question. What were the criteria for applying type 1 or type 3?

Response 6: Thank you for pointing that out. We have explained that in the “Methods” section of our revised manuscript. Due to limited resources, the majority of our patients (86.4%) were subjected to type 3 sleep study, while type 1 sleep study (13.6%) was applied in patients who, by their history, or physical examination, caused high suspicion of suffering from additional sleep disorders, other than OSAHS.

Comments 7: The results present several problems. Tables should not be "broken" between pages. I suggest increasing or formatting the tables so this doesn't happen.

Response 7: Thank you for pointing that out. In our original submission we had presented the tables in this way, however, that changed during the editing of the manuscript, since the journal uses a specific template.

Comments 8: With the sample being 2359 patients – in Table 1, we have a total for Family Status of only 1942 and 700, respectively. What happened to the remaining patients? If they do not have information, they must include an item without information in the table.

Response 8: Thank you for the meticulous revision of our manuscript. As we explained in our answer of your 5th point, some of our patients did not answer all the questions of each questionnaire. For Fagerström scale in particular, the number of the answers is much smaller than the total sample size, as this scale was answered only by those who were current smokers, at the time of their visit. For all the other questionnaires, the number of missing data is relatively small, compared to the total sample size of 2359 patients. In any case, this fact might have caused a participants’ bias, something that we mention in the “limitations” section of our revised manuscript. Furthermore, we would like to point out, that in our original manuscript, we had included the number of patients who had answered in each question, so that every reader, including the reviewers, could have a precise picture of the sample power for each statistical test. It might be more precise to include an item “without information” in each question of each table, in which there were missing data, since the number of missing data was different in each question. However, this not only would have made all the tables much larger, and consequently, more difficult to interpret by the readers, but also would have cause the false impression, that the missing data were included into the statistical analysis, a fact that would be particularly misleading for the readers. Therefore, we avoided to include such an item in each question of each table, however, we point out the existence of missing data not only in the “limitations” section, but also in the “statistical analysis” section of our revised manuscript.

Comments 9: In Table 2 – Do the authors not include the item never in alchohol? The total number of patients does not coincide with the total sample – 2314 and 776.

Response 9: Thank you for pointing that out. It was certainly a mistake not to include the option “never” in the question about alcohol use, however, there was the option “almost never”, which is very similar. Of course, for some people these two are different. In any case, there were relatively few missing data in this question. As far as the second part of your point, it is similar with your 8th point, therefore we refer you to the respective answer of ours.

Comments 10: Table 3 shows missing patients from the sample in the Berlin questionnaire (42 and 5 patients are missing in each column) and in the Stop Bang questionnaire. What happened? If you don't have the data, you must declare it.

Response 10: Thank you for pointing that out. This point is similar with your 8th point; therefore, we refer you to the respective answer of ours.

Comments 11: Regarding Legs movements, there are also patients missing in total from both columns (24 in column 1 and 1 in column 2). What happened?

Response 11: Thank you for your comment. This point is similar with your 8th point; therefore, we refer you to the respective answer of ours.

Comments 12: Table 4 – Dry Mouth patients are missing in both columns: Morning fatigue, heavy Headache, and Headache.

Response 12: Thank you for your comment. This point is similar with your 8th point; therefore, we refer you to the respective answer of ours.

Comments 13: In Bad Mood, in Table 4, sample members are missing. Do they all have a bad mood? There is no answer category "never". The same applies to Snoring loudness, breathing pauses during sleep, and night awakenings. Is there a lack of patients/responses, and has no one ever responded?

Response 13: Thank you for pointing that out. The fact that in many questions the option “never” was not available is certainly a mistake, however the option “almost never”, which is very similar, was available. Perhaps, this is a matter of semantics, since “never” is a very strong term, and when designing the questions it might have been thought extremely unlikely for someone, not to be in a bad mood not once in an entire lifetime. In any case, in the “limitations” section of our revised manuscript, we mention that the design of some questions might have caused a possible bias, since they had not been validated first. As far as the other part of your point, it is similar with your 8th point, therefore we refer you to the respective answer of ours.

Comments 14: The discussion seems short to me, and the limitations should not be in the conclusion but in the discussion.

Response 14: Thank you for your excellent comment. We have changed our revised manuscript accordingly.

Reviewer 2 Report

Comments and Suggestions for Authors

The article "Nicotine dependence alters disease features in patients with OSAHS" offers an original and valuable look into the relationship between nicotine dependence and OSAHS characteristics. With a large dataset of over 2000 patients, the study provides robust statistical findings using appropriate methods, including t-tests, ANOVA, and regression.

A limitation is the reliance on subjective data gathered mainly through questionnaires, which introduces potential recall bias. Additionally, the methods for assessing OSAHS are not fully detailed, and most patients underwent type 3 sleep studies, which are less precise than type 1.

Despite these constraints, the study significantly contributes to understanding the link between nicotine dependence and OSAHS severity with a large, well-analyzed dataset.

Author Response

Comments 1: A limitation is the reliance on subjective data gathered mainly through questionnaires, which introduces potential recall bias.

Response 1: Thank you for pointing that out. We have changed the “limitations” section of our revised manuscript accordingly.

Comments 2: Additionally, the methods for assessing OSAHS are not fully detailed.

Response 2: Thank you for pointing that out. We have made the respective changes in the “methods” section of our revised manuscript.

Comments 3: Most patients underwent type 3 sleep studies, which are less precise than type 1.

Response 3: Thank you for pointing that out. We have changed the “methods” and the “limitations” sections of our revised manuscript accordingly. In the “methods” section we have added the percentage of patients who underwent each type of sleep study and the rational behind that, and in the “limitations” section we have changed the phrasing of this limitation.

Reviewer 3 Report

Comments and Suggestions for Authors

This study examined the relationship between smoking history, nicotine dependence and characteristics of obstructive sleep apnea hypopnea syndrome (OSAHS) as well as other sleep disorders. The authors found significant correlations between tobacco consumption and nicotine dependence with certain characteristics of OSAHS, as well as between nicotine dependence and abnormal sleep behaviors. The topic is interesting as the relationship between smoking and sleep disorders remains controversial. However, this manuscript does not offer new insights regarding the methodology. Furthermore, it lacks clarity and contains considerable overlap of ideas, making it difficult to identify and interpret the relevant results. Below are some specific comments:

The title seems inappropriate because, (1) the study focused not only on the effects of nicotine dependence on sleep and (2) it also examined the relationship between smoking and other sleep disorders in addition to OSAHS.

Introduction

Page 2; line 69-71: How frequent awkenings and sleep architecture fragmentation constitute physiopathological mechanisms by which smoking affects OSAHS.

Page 2; line 85 – 87: The aims of this study are not well justified: What is novel about the methodology that could lead to more relevant findings compared to other studies? Why did the authors choose to examine the relationship between nicotine dependence and OSAHS, as well as other sleep disorders in addition to studying the relationship between smoking history and OSAHS and other sleep disorders?

Methods

Page 2; line 89: Is this really a cross-sectional study? The method of data collection is not mentioned but it appears that the data were collected retrospectively from the records of patients who visited a sleep clinic between September 2010 and September 2023.

Page 2; line 94-95: How was consent for participation in the study obtained?

Page 3; line 98-113: Regarding patient characteristics, the source and/or method of data collection are not mentioned.

Page 3; line 111 – 113: The apnea hypopnea index (AHI) may be underestimated in patients undergoing a type 3 sleep study, which may constitute a bias and a limitation of this study.

Overall, it would have been better to subdivide the methods section into different parts, detailing the recruitment of subjects and classifying the studied characteristics according to their nature (anthropometric, comorbidities, clinical, polygraphic or polysomnographic, etc).

Results:

In order to highlight relevant findings more clearly, the results section should also be divided into different parts, ensuring perfect coherence with the methods section.

Discussion:

The discussion section should be more structured and better aligned with the results. It should also begin by clearly summarizing the most important findings of the study.

The findings regarding the relationships between smoking history and sleep disturbances, as well as  between nicotine dependence and sleep disturbances, were not succinctly compared or interpreted. It would also be valuable to link these findings to potential pathophysiological explanations of the effects of smoking on sleep.

Limitations and study highlights should be addressed in the Discussion section, not in the Conclusion.

Author Response

Comments 1: The title seems inappropriate because, (1) the study focused not only on the effects of nicotine dependence on sleep and (2) it also examined the relationship between smoking and other sleep disorders in addition to OSAHS.

Response 1: Thank you for your suggestion. We have changed the title of our revised manuscript accordingly.

Comments 2: Page 2; line 69-71: How frequent awkenings and sleep architecture fragmentation constitute physiopathological mechanisms by which smoking affects OSAHS.

Response 2: Thank you for your comment which gives us the opportunity to justify that. In the revised version of our manuscript, we have added explanatory phrases for the pathophysiological mechanisms by which, according to existing literature, smoking affects OSAHS. More specifically, it has been shown that smoking causes frequent awakenings and sleep architecture fragmentation during night, which in turn exacerbate daytime sleepiness, which is a key factor of OSAHS.

Comments 3: Page 2; line 85 – 87: The aims of this study are not well justified: What is novel about the methodology that could lead to more relevant findings compared to other studies? Why did the authors choose to examine the relationship between nicotine dependence and OSAHS, as well as other sleep disorders in addition to studying the relationship between smoking history and OSAHS and other sleep disorders?

Response 3: Thank you for pointing that out and give us the opportunity to improve this section of our manuscript. In the “Introduction” section of our revised manuscript, we have justified with more details the aims of this study. We chose to examine the relationship between nicotine dependence and smoking history on the one hand and OSAHS and other sleep disorders on the other, since this relationship is not quite clear, despite the fact that other studies have examined it in the past. What’s more, we believe that our methodology is somehow novel, as it uses quantitative indices not only to investigate for possible relationship between smoking and sleep, but also correlate this this relationship in a quantitative manner.

Comments 4: Page 2; line 89: Is this really a cross-sectional study? The method of data collection is not mentioned but it appears that the data were collected retrospectively from the records of patients who visited a sleep clinic between September 2010 and September 2023.

Response 4: Thank you for your comment which gives us the opportunity to justify that. A cross-sectional study is a type of research design in which someone collects data from many different individuals at a single point in time. In our study, this single point in time was the visit of our patients in our sleep clinic. Despite the fact that this procedure lasted for 13 years, each patient visited our clinic at only a single timepoint during this period. On the contrary, in a longitudinal study, which is the opposite of a cross-sectional study, someone collects data repeatedly from the same subjects over time, often focusing on a smaller group of individuals that are connected by a common trait. However, in our study, we did not collect data from our patients repeatedly over the course of this 13-year period, but only at a single timepoint, the time of their visit. In any case a longitudinal study is much more qualitative than a cross-sectional study for many reasons, thus it would be, somehow “in our interest” to claim that our study was a longitudinal one, i.e. a cohort study, however, in our opinion, that would be neither true nor right.

Comments 5: Page 2; line 94-95: How was consent for participation in the study obtained?

Response 5: Thank you for your comment that gives us the opportunity to explain that. When our patients visit our sleep clinic, they are given a pack of papers with different questions and questionnaires to answer, i.e. the Berlin and STOP bang questionnaires, the Epworth Sleepiness Scale (ESS), the Rosenberg self-esteem scale, the Athens insomnia scale (AIS), the Fagerström scale etc. At the end of this pack of papers, there is a question regarding the patients’ consent for the use of all this information for research purposes. This consent might concern different studies, even studies that have not been designed at the time of consent, a fact that is explained to the patients before they give their consent. We only use the information of the patients that provide this consent and only after obtaining the respective approval by our hospital’s or our university’s ethics committees, which are always aware of the procedure by which we obtained this consent.

Comments 6: Page 3; line 98-113: Regarding patient characteristics, the source and/or method of data collection are not mentioned.

Response 6: Thank you for your excellent comment which gives us the opportunity to improve the quality of our manuscript. We have made all the necessary additions in the “methods” section of our revised manuscript following your suggestion.

Comments 7: Page 3; line 111 – 113: The apnea hypopnea index (AHI) may be underestimated in patients undergoing a type 3 sleep study, which may constitute a bias and a limitation of this study.

Response 7: Thank you for your excellent comment. We have changed the “limitations” section of our revised manuscript accordingly.

Comments 8: Overall, it would have been better to subdivide the methods section into different parts, detailing the recruitment of subjects and classifying the studied characteristics according to their nature (anthropometric, comorbidities, clinical, polygraphic or polysomnographic, etc).

Response 8: Thank you for pointing that out and giving us the opportunity to improve our manuscript. In the revised version of our manuscript, we have changed the “methods” section, following your suggestion.

Comments 9: In order to highlight relevant findings more clearly, the results section should also be divided into different parts, ensuring perfect coherence with the methods section.

Response 9: Thank you for pointing that out. We have changed the “Results” section of our revised manuscript accordingly.

Comments 10: The discussion section should be more structured and better aligned with the results. It should also begin by clearly summarizing the most important findings of the study.

Response 10: Thank you for pointing that out. We have made the respective changes in the “Discussion” section of our revised manuscript.

Comments 11: The findings regarding the relationships between smoking history and sleep disturbances, as well as  between nicotine dependence and sleep disturbances, were not succinctly compared or interpreted. It would also be valuable to link these findings to potential pathophysiological explanations of the effects of smoking on sleep.

Response 11: Thank you for your excellent comment which gives us the opportunity to improve the quality of our manuscript. We have made the respective changes in the “Discussion” section of our revised manuscript.

Comments 12: Limitations and study highlights should be addressed in the Discussion section, not in the Conclusion.

Response 12: Thank you for pointing that out. We have changed the respective sections of our revised manuscript accordingly.

Round 2

Reviewer 1 Report

Comments and Suggestions for Authors

The authors made most of the corrections to clarify some study design limitations and thus facilitate its understanding. The division by topics makes it easier to read. However, the tables and titles need to be revised. The tables cannot be divided by pages, which makes them difficult to analyze. Consider dividing them or reformatting them, so they occupy only one page. 

Thank you.

Author Response

Comments 1: The authors made most of the corrections to clarify some study design limitations and thus facilitate its understanding. The division by topics makes it easier to read. However, the tables and titles need to be revised. The tables cannot be divided by pages, which makes them difficult to analyze. Consider dividing them or reformatting them, so they occupy only one page. 

Response 1: Thank you for your comment. Following your advice, we have changed the titles of our tables, so that they would better reflect the topics of our manuscript, and we have also reformed and divided our tables, in order to occupy only one page, so that they would be easier to analyze.

Reviewer 3 Report

Comments and Suggestions for Authors

I would like to thank the authors for addressing most of my concerns. This revised version of the manuscript has significantly improved and I consider it suitable for publication.

Author Response

Comments 1: I would like to thank the authors for addressing most of my concerns. This revised version of the manuscript has significantly improved and I consider it suitable for publication.

Response 1: Thank you